# Motivators and Barriers to Physical Activity among Youth with Sickle Cell Disease: Brief Review

**DOI:** 10.3390/children9040572

**Published:** 2022-04-17

**Authors:** Olalekan Olatokunbo Olorunyomi, Robert Ie Liem, Lewis Li-yen Hsu

**Affiliations:** 1Division of Pediatric Hematology-Oncology, University of Illinois at Chicago, Chicago, IL 60612, USA; olalekanol@pcom.edu; 2Division of Hematology, Oncology & Stem Cell Transplant, Ann & Robert H. Lurie Children’s Hospital of Chicago, Chicago, IL 60611, USA; rliem@luriechildrens.org

**Keywords:** behaviors, exercise, facilitators, obstacles, adolescents, self-efficacy

## Abstract

Purpose: Health disparities for minority groups include a low rate of physical activity and underserved urban minority youth with chronic disease are among the least active population segments, as exemplified by sickle cell disease (SCD). “Exercise prescriptions” for youth with chronic diseases need to be evidence based and align with psychologic motivators and barriers. This scoping review sought evidence for psychosocial motivators or barriers to physical activity (PA) in youth with SCD and other chronic disease that could be relevant to SCD. Methods: Five databases were searched for studies on urban minority youth published between 2009 and 2022. Results: Keyword searching yielded no papers on SCD and PA motivation and barriers. Adding health-related quality of life (HRQL) in SCD found eleven relevant papers. Widening the search to chronic disease in minority youth resulted in a total of 49 papers. Three thematic categories and seven sub-themes emerged. PA barriers added by chronic disease include fear of triggering disease complications, negative relationships due to disease limitations on performance in sports, and lack of suitable environment for PA that accommodates the chronic disease. PA motivators are similar for youth without chronic disease: self-efficacy, autonomy, positive relationships with peers and parents and coach/teacher. Conclusion: Direct descriptions of PA motivations and barriers to PA in SCD are limited to fatigue and fear of sickle vaso-occlusive pain. The PA barriers and motivators found for urban youth with chronic disease overlap with themes in healthy adolescents from underserved minorities. Community-based interventions could strengthen PA motivators (self-efficacy, autonomy, positive relationships with peers and parents and coach/teacher) but need disease accommodations to overcome the barriers (fear of triggering disease complications, environmental limitations, and negative relationships). Evidence-based exercise prescriptions might incorporate educational modules to overcome disease stigma and misconceptions. Prospective studies of PA motivators and barriers could improve HRQL in SCD.

## 1. Introduction

The American College of Sports Medicine (ACSM) and World Health Organization (WHO) recommend that children have at least 60 min of moderate to vigorous-intensity physical activity (MVPA) daily, but implementation has met many barriers especially among those with chronic disease. Promoting physical activity (PA) in adolescence is emphasized by physicians [1] because healthy lifestyle habits are formed in this crucial stage of development [2,3]. However, the 2017 Youth Risk Survey reported that only 26.1% of healthy American children met this baseline recommendation [4]. Recently, there has been a growing call to implement PA assessments and counseling for children in clinical settings, and specific evidence-based exercise prescriptions for children and youth with special health care needs (CYSHCN) [1,5,6], including pediatric chronic cardiorespiratory disease [7]. Furthermore, it is widely recognized that subgroups of children are likely to have even lower levels of PA than the general population: youth in ethnic minority groups and adolescents [1,8,9]. PA disparities for adults from underserved minority groups have been recently reviewed and best practices have been identified: community-based participatory research (CBPR), multi-level interventions, and interdisciplinary teams [10].

Consideration of the barriers to PA for sickle cell disease (SCD) can illustrate why a clinician might need to provide specific “exercise prescription” for underserved minority adolescents with a chronic disease. SCD is an inherited blood disorder that disproportionately affects individuals of African or Afro-Caribbean descent. In a recent global survey, over half of 1500 SCD patients responded that they avoid PA because of concerns about pain, exhaustion, or dehydration [11]. Fatigue is the most common daily impairment to quality of life reported by individuals with SCD to limit activities of daily living as well as endurance in PA [11]. In a survey of 100 children with SCD, 90% reported participation in physical education classes but only 48% participation in sports, and their participation in moderate-to-vigorous PA is for shorter durations of time [12]. SCD is associated with unsubstantiated concerns that physical overexertion could trigger disease-related complications because of the pro-inflammatory nature of the disease [13]. SCD is associated with decreased cardiorespiratory fitness and abnormal cardiorespiratory responses to exercise [14]. In addition, some youth with SCD have functional deficits from sequelae of stroke or avascular necrosis of the hip that could also limit PA and exercise. Finally, SCD patients also have disproportionately further risk factors for low PA: disadvantages of low income and urban residence [11]. In favor of exercise, selected groups of adults and adolescents with SCD have successfully tolerated maximal exercise testing and the 6-min walk protocol [15,16,17] and exercise training programs [18]. Children and adolescents with SCD can receive mixed messages from physicians about participation in PA [13,19] and there are currently no formal evidenced-based guidelines to support PA recommendations for the SCD population [20].

Thus, although experts strongly encourage PA for adolescents [1], the factors that are associated with good adherence to prescribed exercise remain unclear. Adherence to a PA program may be improved when the recommended PA is a good match for the individual’s motivation and addresses barriers to PA for that individual.

We therefore reviewed the scientific literature with a focus on the psychosocial motivators and barriers to PA behaviors in children and adolescents with SCD and other chronic diseases. So that “exercise prescriptions” will be successful, a deeper understanding of these motivators and barriers could be useful for building effective interventions to promote PA among youth with SCD.

## 2. Methods

This literature review examined the research question: what are the motivators and barriers to PA among youth with chronic disease? The review was limited to children with chronic disease from racial/ethnic minorities, including adolescents with SCD. The following key words and phrases were searched in the Web of Science (WOS), PubMed, CINAHL, Psych Info, and Google Scholar databases: motivation *, barriers *, physical activity OR sport * OR exercise OR health-related quality of life *, child * OR adolescent * OR pediatric *, facilitator *, aversion *, physical activity behavior *, attitudes and beliefs *, physical disability *, chronic disease OR illness *, and sickle cell disease. The inclusion criteria were English articles published from January 2009 to March 2022. Articles were included if they included children and adolescents with chronic disease from racial/ethnic minorities. Health-related quality of life (HRQL) was included because physical activity and physical function is included in many pediatric HRQL measures [21]. Titles and abstracts were screened and articles that meet initial screening were organized in the citation manager. The appropriateness of included articles was confirmed by a second reviewer. After final selection, references from included articles were reviewed and additional appropriate studies that were missed through the initial search were added.

Thematic analysis was conducted based on social ecologic theory according to the approach of Thomas and Harden [22]. The social ecological model is a framework for understanding the effects of personal, interpersonal, environmental, and policy factors that determine behavior [23]. Using the abductive approach to thematic synthesis, we followed the stages of familiarization, generated initial codes, organized codes into related areas to construct descriptive themes and sub-themes, developed analytical themes based on the chosen theoretical framework and identified key themes and sub-themes.

## 3. Results

The search for articles on psychosocial factors that influence PA in youth with SCD yielded no papers that focused on motivation and barriers to physical activity. One study focusing on perception of fatigue when approaching the anaerobic exercise threshold found that 73% of interviewees with SCD did not stop exercising until symptoms were severe (chest tightness, blurry vision) [20]. Ten papers on health-related quality of life (HRQL) in SCD mentioned physical function, but only two had relevance to psychosocial factors that influence PA. One study determined that youth on hydroxyurea for SCD had better HRQL, including improved physical activity [24]. The other paper examined the family impact of SCD and noted that parents and siblings of teens with SCD mentioned physical activity limitations as one aspect of chronic disease impact on the family’s quality of life, with an indirect mention of weaker bones as a barrier to PA [25]. These descriptions were not sufficient for thematic analysis of psychosocial factors and PA in SCD.

Because thematic analysis was not possible with the studies on SCD, the search was expanded beyond SCD to the broader population of “chronic disease in urban minority youth” to provide more material for qualitative analysis. This broader search resulted in 118 relevant citations by title and abstract. Review of the full texts narrowed the group, then a manual search added 10 studies to result in 38 studies on PA motivators and barriers in youth with other chronic diseases (Figure 1) in addition to the 11 studies on HRQL in SCD.

Review of the references from these studies in chronic diseases other than SCD led to seven additional papers that were considered relevant for inclusion in qualitative analysis (Figure 1). Thematic analysis of motivators and barriers resulted in three categories of the social ecological model (personal, interpersonal relationships, and environment) and seven sub-themes (Table 1). No information was found in the policy category of the social ecological model.

### 3.1. Personal

Self-Efficacy—Under the personal category, self-efficacy was a frequent and broad term used in the studies reviewed [26]. Adolescence is a developmental stage characterized by rapid physical and emotional changes that increase the demand on an individual to respond or change. Adolescents with chronic illnesses or disabilities lack confidence in their ability to participate in PA, exercise, or sports [27]. Increased fatigability is a common feature of SCD [11,12,13,14,15,16,20] that may limit participation in endurance sports and have negative consequences on self-perception. Children with other chronic diseases, such as myalgia from encephalomyelitis, experience lower self-confidence and increased sense of vulnerability due to the physically restrictive nature of their illness [28]. In adolescents with obesity, low self-efficacy is often reported and results in decreased exercise activity.

Fear—Fear of triggering vaso-occlusive pain with over-exertion limits PA in SCD, and the extreme intensity of sickle vaso-occlusive pain can make this fear significantly impair HRQL in SCD [24]. Fear of triggering pain with exercise has caused avoidance behavior in adolescents with chronic illness such as juvenile idiopathic arthritis [29]. Pain intensity and pain duration are likely associated with developing fear avoidance belief, which could result in lower levels of physical activity. Reducing one’s physical activity levels to avoid pain may lead to heighted fear of movement, depression, and worsening of chronic pain [30]. In older adults, pain intensity and duration have been identified to be predictors of self-reported and performance-reported disabilities [31]. In other chronic diseases, disease complications triggered by exercise can potentially be severe and create barriers to PA among adolescents. For example, adolescents with type 1 diabetes and their parents fear the risk of hypoglycemic episodes after participation in PA or exercise [32]. Knowledge and ability gaps due to chronic diseases may be associated with fear and lower self-esteem, leading to limited PA participation and poor development of skills compared to peers, further diminishing self-esteem [27].

Autonomy—Youth prefer to choose how they would like to be active and not have activities chosen for them. Having a choice of PA may represent both a motivator or barrier to PA. Adolescents with obesity perceived non-structured PA lower in priority and unnecessary as they developed and transitioned into young adulthood because they did not enjoy it [33]. However, self-efficacy and goal setting were positively correlated with vigorous PA in youth with multiple sclerosis [34]. Pre-teens with congenital heart disease in a program of structured PA emphasized their enjoyment of PA as a primary source of motivation [35].

### 3.2. Relationships

Interpersonal relationships as a thematic category include the relationships between adolescents and their peers, friends, or family members. Encouragement and role modeling by these individuals may be important factors for increasing PA among sedentary adolescents.

### 3.3. Parental Relationship

Lack of support from parents, who themselves may be inactive, may contribute to decreased motivation for PA among adolescents with obesity [33,36,37]. Parental overprotectiveness may be associated with fear avoidance behaviors that result in reduced PA in adolescents with chronic illness. For example, parental fear may include catastrophizing pain that may or may not occur with PA in children with chronic pain conditions [38]. Some programs describe increased PA when mothers and children exercise together [39,40]. A study found that children with chronic disease had increased adherence to daily PA when family members were engaged [41].

### 3.4. Peer Relationships

For some children with physical disabilities, peer involvement and acceptance are effective motivators of their participation in PA [27]. Playing a game with peers can influence social, emotional, and cognitive development. Many after-school programs promote PA by combining team sports with adult mentorship. However, adverse interactions with peers (e.g., stigma, bullying, social rejection, and isolation) are common experiences for children with chronic diseases [42]. These adverse interpersonal factors may represent significant barriers to active sports participation during spontaneous play or structured training.

Research investigating the effects of active video games (AVG) or exergames have increased over the past 30 years. In the United States, an estimated 91% of children play video games with the population of gamers increasing worldwide [43]. Video games can include significant social interaction among teenagers [44] and cooperative video games can improve child behavior and emotion [45,46]. Exergaming may have some advantages for youth with chronic diseases, including increasing motivation and allowing for immersion and enjoyment [47,48]. Exergaming may also give youth with chronic diseases a chance to increase their PA through non-traditional forms of exercise [49]. Lastly, exergames or AVG’s may be modified or adapted to fit the needs of different populations, which may be beneficial for youth with chronic diseases [50,51].

Studies suggest that professional gaming (esport) may be physically taxing on the body, as supported by elevated heart rates and cortisol levels comparable to that observed in other sports [43]. Adolescents with chronic diseases may be motivated to increase PA in order to improve their performance in video games with their peers. Therefore, some suggest video gaming as an intervention to increase PA in children with chronic illness [42].

### 3.5. Coach/Teacher Relationships

Coaches, instructors, and physical education teachers are key facilitators of PA because they can modify activities to accommodate specific chronic medical conditions [27]. A parent from an interview stated this about the coaches, instructors, and physical education teachers, “A lot comes down to the compassion of coaches. If they understand the situation then they give them the attention they (children) need”. However, some adult coaches, teachers, or mentors might have low awareness of the accommodations needed for adolescents with chronic illness. A physically disabled adolescent in a study stated that “Some teachers I’ve had in PE they don’t want to like really listen. They say we’re doing it this way and that’s it, you have to adapt to try and fit in” [27].

## 4. Environment

Environment as a thematic category includes safe and accessible environments to participate in PA. Environmental barriers include limited open or crowded space for play or sports participation; lack of safe play space, which is common in urban settings; or limited access to equipment or accommodations required by children with chronic disease (e.g., swimming pool that is heated or accessible for physical disability). Exposure to harsh climates may limit PA for some children with chronic diseases such as SCD, for whom extreme heat or cold may trigger complications. In one study, 41% of 234 adolescents with disabilities stated that uneven playgrounds were a barrier to PA [52]. The ability or inability to adapt PA programs and activities or provide access to transportation was frequently reported as a barrier or facilitator of PA [53,54,55]. 

## 5. Discussion

Evidence is slowly accumulating for customized physical activity prescriptions in youth with SCD and other chronic diseases [7,13,56], but these prescriptions would be enhanced by incorporating strategies to overcome behavioral barriers to PA in youth with SCD and other chronic diseases. This literature review suggests that youth with SCD have some barriers and motivators to PA that are unique to SCD (Table 2). However, the literature remains incomplete, and none could be included in Table 1. For example, this search strategy found nothing on avascular necrosis (AVN) of the femoral head as a PA barrier for adolescents [25]. AVN affects as many as 50% of adults with SCD and some adolescents, and a body of orthopedic studies use limited mobility and poor HRQL as outcome measures to compare total hip arthroplasty techniques for SCD AVN [57].

Thus, to learn more about possible PA motivators and barriers in SCD, this study examined the PA motivators and barriers in other under-resourced youth with chronic diseases, which are also outlined in Table 2. Youth with chronic diseases share in three broad categories of perceived barriers or motivators of PA. Personal factors include self-efficacy, fear, and autonomy. Interpersonal factors include the relationship of children with chronic disease to their parents, peers, and coaches/teachers. Environmental factors may refer to available adaptations or accommodations required for children with chronic disease to engage in PA. Although it is unclear which factors may contribute most to PA participation in adolescents with chronic disease, much attention has focused on self-efficacy as one of the more important motivators or barriers to PA in this population.

In the example of SCD, youth with SCD are encouraged by health providers to participate in PA and to take advantage of the physical and psychological benefits it offers. Untrained youth might allow high motivation to override judgment about when to stop exercising [20]. However, there are unsubstantiated concerns that PA may trigger complications of SCD, whether pain or other complications. SCD is known to decrease cardiopulmonary fitness because of its pathophysiology [14]. Gouraud [58] observed that patients with SCD (hemoglobin SS type) had a lower step frequency on a 6-min walk test compared to healthy control subjects and hypothesized that SS patients walked more slowly to decrease the metabolic cost and perceived fatigue. An exercise training feasibility study found teens with SCD showed a 10 percent increase in their fitness after 6 weeks of aerobics training and 12 weeks of a home exercise program [56]. However, during the second part of the study, teens showed decreased adherence to the exercise program with some expressing decreased interest and motivation. Despite the robust literature on the physiological contributors to reduced fitness in youth with SCD, few studies have examined the psychosocial factors that affect PA in this population.

In a study that examined PA patterns in children with SCD compared to the national control sample, negative perceptions related to SCD were correlated with decreased PA levels in children with SCD [12]. The themes that might be relevant to youth with SCD are self-efficacy, fear, and parental relationship. Effects of SCD such as delayed development are observed in youth with SCD, so they are likely to be smaller than their peers. Physiological impairments may also decrease their belief in their personal ability, which in turn affects their self-efficacy. Fear might be a determinant of PA levels in SCD because youth with SCD are likely to experience fear and avoid activities that may trigger pain episodes. Parental relationships could influence PA levels because parents may exhibit overprotection of their children to avoid triggering complications of their children‘s disease.

A recent review of PA disparities for adults from underserved minority groups identified community-based interventions as “best practices” that overcome some environmental barriers to PA: community-based participatory research (CBPR), multi-level interventions, and interdisciplinary teams [10]. Adapting these community interventions to adolescents with chronic disease will be a challenge, because of dependence on interpersonal relationships. Rare diseases such as SCD are associated with community stigma or misunderstanding [60,61], and so the very community members that promote PA could also hinder the motivation for PA in the adolescent with chronic disease. Therefore, implementing these “best practices” will probably require careful addition of medical interaction, emphasizing the interdisciplinary team.

This review has several limitations. The body of literature on the motivations and barriers to PA in youth with chronic disease is small. Although the primary interest of this review is SCD, no studies in SCD were found to be designed to study motivations and barriers to PA. The SCD findings in Table 2 are drawn from seven studies with motivations and barriers as secondary results. The study thus searched for primary evidence on populations related to SCD, minority youth with other chronic disease, because youth with SCD are racial and ethnic minorities in North America and Europe. However, the search on minorities excludes youth with some chronic diseases that do not affect minority groups, notably cystic fibrosis. Gaps in knowledge certainly remain after this literature search but understanding motivations and barriers may lead to ways to increase PA participation in children with chronic disease such as SCD. Other aspects that require clarification include the following: which factors are most important to improve PA, which options should be included in a prescription to increase PA, which strategies for PA should be matched to which chronic disease (especially SCD), and what are the levels of incentive and reinforcement that will sustain high PA during a lifetime that is affected by chronic disease? How much adaptation is needed for individual personalities, local environments, and community contexts?

Youth with chronic disease may miss out on the spectrum of psychological and physiological benefits of PA due to a lack of appreciation for their motivations and barriers. A better, more thorough understanding of these factors may provide additional insight for developing appropriate evidence-based guidelines for increasing PA and exercise in the SCD population. Knowledge gaps on these motivations and barriers should be filled so that exercise prescriptions can be implemented for youth with chronic disease such as SCD. Prospective studies on the physiologic and psychosocial factors in PA in SCD are vitally important to improve the quality of life in SCD.

## Figures and Tables

**Figure 1 children-09-00572-f001:**
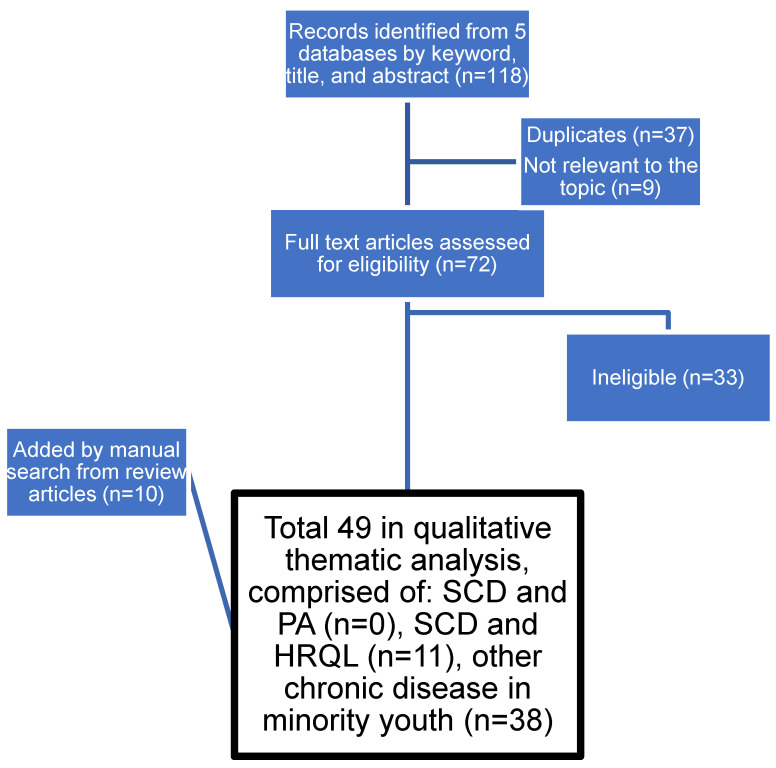
PRISMA flowchart of the literature search and screening.

**Table 1 children-09-00572-t001:** Thematic analysis results using social ecological model.

Psychosocial Factors	Motivation Theme	Barrier Theme
Personal		
Self-Efficacy = Intrinsic factor that emphasizes a person’s belief in their abilities.	Youth with higher self-efficacy may likely be motivated to participate in physical activity (PA).	Youth with lower self-efficacy might experience lower motivation to participate in PA, especially group activities. Youth who had less opportunity to practice a sport feel that their skills are below peers.
Autonomy = A person’s ability to make choices for themselves among PA options.	Autonomy and variety provide opportunity for youth to exercise independence in deciding among options. This may result in motivation in youth to participate in PA.	The lack of options or freedom to exercise independence may decrease motivation in youth participation in PA.
Fear = Fear of injury or risk of adverse side effects from PA.		Fear may decrease motivation to participate in PA for youth with chronic diseases. Youth may avoid PA to prevent triggering disease complication such as pain or exhaustion.
Relationships		
Parental = Parents can encourage PA or even share in PA with the youth.	Parental cooperation, support, and encouragement may increase motivation to participate in PA.	Over protection and diminished parental cooperation may likely decrease motivation to participate in PA.
Peer = PA with peers can be powerful as a motivator or as a barrier.	Social Inclusion, Peer support and comradery in team settings may provide encouragement and likely increase motivation to participate in PA.	Bullying and teasing may decrease motivation to participate in PA.
Coach/Teacher = Youth spend approximately 7 h. daily at school excluding afterschool programs.	Coach/ Teacher involvement could be another source of support and encouragement. Professionals who work with youth could be trained to modify activities to involve youth with chronic disease, which may lead to increase motivation to participate in PA.	Inexperienced coach/teachers may decrease the motivation of youth for PA participation. This may prove to be a barrier due to the inability to effectively modify activity for youth or speak motivationally to these youth with chronic disease. Over protection in these interactions may also decrease motivation to participate in PA.
Environment		
Environmental = Safe space for PA, trained supervision, and helpful equipment makes for a good environment for PA.	Availability of resources at home, school, or community may increase motivation to participate in PA.	The lack of resources at home, school, and community may decrease motivation to participate in PA. Urban environments and poverty can severely limit safety and space for PA, creating barriers and decreasing motivation to participate in PA.

PA = physical activity.

**Table 2 children-09-00572-t002:** Physical activity motivators and barriers for youth with sickle cell disease.

	Barriers	Refs	Motivators	Refs
Sickle cell disease specific	Fatigue Lower threshold for anaerobic exercise Exertion triggering vaso-occlusive pain Avascular necrosis of femoral head	[11,20,25,57,58]	Hydroxyurea treatment can improve exercise tolerance	[24,59]
Youth	Peer relationships might not support PA Activities not matched for age Lack of fun Low perception of competence	[19,42]	Peer support for PA Mentor/coach/family support Having fun	[25,34,35,39,40,41,42]
Under-resourced	Inadequate facilities Unsafe urban environment Competing priorities for time Cultural expectations	[52,53,54,55]	Community-based interventions	[10]
Chronic disease	Stigma, performance hampered by chronic disease Worry that PA will cause chronic disease to worsen Mentors/coaches/family misinformation about disease limitations	[27,28,29,30,32,38,60,61]	Reassurance from medical personnel Evidence that PA can be safe Tailored PA accommodates for chronic disease	[7,13,18,56]

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
