# Peer review of "Motivators and Barriers to Physical Activity among Youth with Sickle Cell Disease: Brief Review"

_children, 2022, doi:10.3390/children9040572_

Round 1

Reviewer 1 Report

Even though this brief review of the literature sounds interesting, it seems incomplete. There are some important points that need to be clarified before publication.
What are the authors trying to convey, motivators and barriers to physical activity among young people with chronic diseases in general or with sickle cell disease (SCD)? Depending on the objective of this literature review, the discussion should be improved, since it seems that the target population is SCD. If the target population is SCD, I would like to suggest the title "Motivators and barriers to physical activity among youth with sickle cell disease: A brief review" and go a little deeper into this particular disease. Yes, the target population is adolescents with chronic diseases, the methodology should be improved since the authors do not mention that they have done a search by type of chronic disease in this age group. This should be mentioned as a limitation of the study, in the limitations and suggestions paragraph. Likewise, another important limitation of this study is the type of population studied (racial/ethnic minorities from North America and the United Kingdom), which would make it difficult to generalize to other adolescents with CFD. Table 1 appears in the discussion, which is not referred to at any time in the manuscript. The entire manuscript must be improved before publication.

Minor points are shown in the manuscript.

Reviewer 2 Report

Thank you for the opportunity to get acquainted with the work.The article is interesting and may have a practical application. The authors should supplement it with the Health-Related Quality of Life. The use of Health-Related Quality of Life to assess activity in a patient with chronic diseases enables a broader view. Literature should be supplemented with the quality of life among others Lechosław Paweł Chmielik , Grażyna Mielnik-Niedzielska , Anna Kasprzyk 1, Tomasz Stankiewicz ,Artur Niedzielski .Health-Related Quality of Life Assessed in Children with Chronic Rhinitis and Sinusitis ,Children 2021, 8(12), 1133; https://doi.org/10.3390/children8121133

Round 2

Reviewer 1 Report

The quality of this manuscript has improved by focusing on SCD. However, in various parts of the manuscript, they do not emphasize that the study is primarily focused on this disease. Although the authors have been able to add more studies, the body of the article has not changed ¿ why? I suggest that the authors emphasize the importance of knowing the motivating factors and barriers that affect physical activity among young people with SCD and the need to carry out prospective observational studies that consider these factors of vital importance to improve their quality of life. I have not found tables 1 and 2 mentioned by the authors and would like to see them.

Author Response

Open Review – Review#1 round 2

The quality of this manuscript has improved by focusing on SCD. However, in various parts of the manuscript, they do not emphasize that the study is primarily focused on this disease.

Added this point in Results (lines 130-137) and Discussion (lines 282-283, lines 334-336, line 346).

Although the authors have been able to add more studies, the body of the article has not changed ¿ why?

Added to the section on Self-efficacy (line 161-162) and Fear (lines 166-168) with rearrangement and revision of the rest of the paragraph on Fear to improve the flow of ideas (lines 168-201)

added Result and Discussion about very scant mention of avascular necrosis of the hip (line 128, lines 276-281 and Table 2

clarified that no SCD motivators and barriers had enough detail to include in Table 1. However, 7 papers had PA motivators and barriers as secondary data in SCD and these are in Table 2 (line 276-277

I suggest that the authors emphasize the importance of knowing the motivating factors and barriers that affect physical activity among young people with SCD and the need to carry out prospective observational studies that consider these factors of vital importance to improve their quality of life.

Thank you for this suggestion. Added this point to the abstract (lines30-31) and the Discussion (lines 356-357)

I have not found tables 1 and 2 mentioned by the authors and would like to see them.

We apologize that an uploading problem did not include Tables 1 and 2 in one version of the manuscript. Tables 1 and 2 are in the latest manuscript version.

Reviewer 2 Report

Accept in present form

Author Response

Thank you.